# Clinical Efficacy of Carbocysteine in COPD: Beyond the Mucolytic Action

**DOI:** 10.3390/pharmaceutics14061261

**Published:** 2022-06-14

**Authors:** Elisabetta Pace, Isa Cerveri, Donato Lacedonia, Gregorino Paone, Alessandro Sanduzzi Zamparelli, Rossella Sorbo, Marcello Allegretti, Luigi Lanata, Francesco Scaglione

**Affiliations:** 1Institute of Translational Pharmacology (IFT), National Research Council, Via Ugo la Malfa, 153, 90146 Palermo, Italy; elisabetta.pace@ift.cnr.it; 2Department of Internal Medicine and Medical Therapy, University of Pavia, 27100 Pavia, Italy; icerveri@smatteo.pv.it; 3Institute of Respiratory Diseases, Department of Medical and Surgical Sciences, University of Foggia, 71122 Foggia, Italy; donato.lacedonia@unifg.it; 4Department of Cardiovascular and Respiratory Sciences, Sapienza University of Rome, 00185 Rome, Italy; rpaone1023@yahoo.com; 5UOC Pneumotisiologia, Scuola di Specializzazione in Malattie Respiratorie, Università degli Studi di Napoli Federico II A.O.R.N. Monaldi-Cotugno-CTO Piazzale Ettore Ruggieri, 80138 Napoli, Italy; alessandro.sanduzzi@unina.it; 6Dompé Farmaceutici SpA, 20122 Milan, Italy; rossella.sorbo@dompe.com (R.S.); marcello.allegretti@dompe.com (M.A.); luigi.lanata@dompe.com (L.L.); 7Department of Oncology and Onco-Hematology, University of Milan, 20122 Milan, Italy

**Keywords:** carbocysteine, carbocysteine pharmacology, carbocysteine human diseases, carbocysteine lung diseases, carbocysteine COPD

## Abstract

Chronic obstructive pulmonary disease (COPD) is a heterogeneous disease with a versatile and complicated profile, being the fourth most common single cause of death worldwide. Several research groups have been trying to identify possible therapeutic approaches to treat COPD, such as the use of mucoactive drugs, which include carbocysteine. However, their role in the treatment of patients suffering from COPD remains controversial due to COPD’s multifaceted profile. In the present review, 72 articles, published in peer-reviewed journals with high impact factors, are analyzed in order to provide significant insight and increase the knowledge about COPD considering the important contribution of carbocysteine in reducing exacerbations via multiple mechanisms. Carbocysteine is in fact able to modulate mucins and ciliary functions, and to counteract viral and bacterial infections as well as oxidative stress, offering cytoprotective effects. Furthermore, carbocysteine improves steroid responsiveness and exerts anti-inflammatory activity. This analysis demonstrates that the use of carbocysteine in COPD patients represents a well-tolerated treatment with a favorable safety profile, and might contribute to a better quality of life for patients suffering from this serious illness.

## 1. Introduction

For the past 20 years, extensive research has been devoted to the pathogenesis and treatment of chronic obstructive pulmonary disease (COPD). Based on clinical presentation, genetic background, pathophysiology, and therapeutic response, COPD is classified as a heterogeneous disease with a versatile and complicated profile. It is crucial to search for precision medicine to treat this pathology [1], which is estimated to be the third most common single cause of death worldwide [2], constituting a major public health problem [3].

According to several studies reported throughout the years, the pathogenesis of COPD has been associated with airway inflammation, oxidative stress, bacteria colonization, cilium-beating dysfunction, and mucus hypersecretion. In particular, a review from 2010 stressed the importance of chronic mucus hypersecretion—the most typical symptom of chronic bronchitis [4,5]—which is present independently of airflow obstruction [6]. Moreover, due to its heterogeneity, it has been challenging to precisely distinguish COPD subtypes to choose the most appropriate treatment for each patient. In the last 10 years, this point has often been reiterated along with the message that the prevention, diagnosis, and treatment of COPD should be a long-term, comprehensive, persistent, and individualized program.

Although mucoactive drugs have been widely used in clinical practice for a long time, and their efficacy has been extensively addressed in the scientific literature, their role in the treatment of patients suffering from COPD remains controversial. There are numerous available mucoactive medicines for prescription; however, it is important to select them properly according to the disease in question [1].

COPD patients with frequent exacerbations experience the recurrence of episodes characterized by an acute worsening of respiratory symptoms that requires additional therapy. Indeed, the recurrence of these episodes accelerates the loss of lung function and leads to decreased health-related quality of life, significant economic costs, and increased mortality rates [7].

Mucoactive agents have been used in a large range of pathologies due to their recognizable potential utility [8]. In 2006, Poole and Black [5] published a systematic review of 26 randomized, placebo-controlled studies that recruited more than 7300 patients with COPD. In this review, it was shown that treatment with mucolytics for at least 2 months significantly reduced the number of exacerbations. Since then, other reviews have been published in the Cochrane Library evaluating the effectiveness of mucolytic agents in the prevention of exacerbations [9]. The correlations found in the studies produced controversial insights, and none of them was able to verify the effect of mucolytics over a prolonged period [10]. The evidence gathered was not precise enough to identify the target population for mucolytic/antioxidant agents in COPD, as the randomized control trials (RCTs) had different quality standards, and the durations and dosages of treatment as well as the number of exacerbations in the year prior to enrolment also differed between the studies [8,10]. As a result, the national guidelines for COPD management in Europe reported different recommendations. N-acetylcysteine and oral carbocysteine were both recommended in the Czech Republic, England and Wales (with caution), Poland, Russia, and Spain, but not recommended in Finland, France, and Portugal [11].

The 2017 report of the Global Initiative for Chronic Obstructive Lung Disease (GOLD) [12] presented a table entitled “Anti-inflammatory therapy in stable COPD”. This table stated that regular treatment with mucolytic/antioxidant agents such as carbocysteine, N-acetylcysteine (NAC), and erdosteine modestly improved health status by reducing the risk of exacerbations in selected populations not receiving inhaled corticosteroids (ICSs). This statement was classified as “evidence B”. According to the description of levels of evidence, sources of “evidence B” are RCTs with important limitations, as these studies include only a limited number of patients, post hoc or subgroup analysis of RCTs, or meta-analysis of RCTs. Moreover, the body of evidence is limited because of important restrictions (e.g., methodological flaws, small numbers, short duration), having been undertaken in a population that differs from the recommended target population, or having results that are in some way inconsistent. The last GOLD report, released in 2020 [13], confirmed previous statements indicating that the evidence should still be classified as B. Due to the heterogeneity of studied populations, treatment dosages, and concomitant treatments, currently available data do not allow the precise identification of potential target populations for mucoactive agents in COPD. Thus, GOLD recommends giving mucolytic and antioxidant agents (e.g., N-acetylcysteine, carbocysteine, and erdosteine) to treat COPD patients who are not receiving inhaled corticosteroid therapy. This suggestion could help to reduce COPD exacerbations and lead to a small improvement in the health status of these patients.

Due to the heterogeneity detected in the studies available concerning COPD and possible mucoactive therapies, this has become a very controversial topic. Recently, a group of Chinese pulmonary physicians worked together to reach a consensus based on a literature review to define guidelines on mucoactive treatment for COPD. The specialists in the field concluded that, in general, mucoactive agents play an important role in COPD, without any association with an increase in adverse effects. However, they repeated once again that precise treatments in the targeted COPD population need further investigation, with a stratification strategy [1]. Moreover, to date, no head-to-head clinical studies on the same COPD patients across different mucoactive agents have been conducted to directly compare their efficacy profiles. It is also important to take into consideration that the cellular and molecular mechanisms underlying the clinical efficacy of mucolytics are largely unknown.

The present review provides a comprehensive vision regarding the clinical efficacy of a mucoactive drug—carbocysteine—and insights into its cellular and molecular activities that could explain its clinical efficacy.

## 2. Materials and Methods

The EMBASE, Medline, and Cochrane review databases were used to conduct a systematic literature search. This search was also extended to grey literature sources, such as abstracts from the Congress of the European Respiratory Society (journal) or the American Thoracic Society (ATS) Journal. The keywords applied for the investigation were carbocysteine, carbocysteine pharmacology, carbocysteine human diseases, carbocysteine lung diseases, and carbocysteine COPD. Throughout the exploration, English-language scientific papers and literature reviews on the molecular and cellular mechanisms of carbocysteine and the clinical efficacy of carbocysteine in patients with COPD were also identified.

No time limit or exclusion criteria were set for the search. All reported molecular and clinical outcomes were evaluated, with no restrictions on the type or stage of publication, sample type, or trial design (RCTs, as well as open, controlled, and uncontrolled trials, were included in the evaluation).

The initial literature search identified 363 published articles, 110 of which were potentially relevant references. The main topics were critically evaluated during the discussion, and after the meeting each expert carried out an independent qualitative systematic literature review for the topic assigned. The references in the identified articles were reviewed for additional sources. Eventually, 80 articles were eligible for this review after group discussion; disagreements were resolved by consensus and, ultimately, 72 articles were included in the review (Figure 1). Considering the nature of this work, for the sake of brevity, we do not list or provide specific comments on the exclusion criteria of papers that were judged by the panel of experts to be less relevant or not strictly related.

## 3. Molecular Effects of Carbocysteine

Carbocysteine (R-2-amino-3[(carboxymethyl)thiol] propionic acid) is a biologically active dibasic amino acid. The carbocysteine molecule (Figure 2) is characterized by the presence of a bound sulfhydrilic group.

The structure and mechanism of action of carbocysteine differ from those of other commonly available mucolytic drugs. For instance, N-acetylcysteine (NAC) and erdosteine bear free sulfhydryl (thiol) groups that allow them to split glycoprotein bonds in mucus [14]. Conversely, a study using animal models demonstrated that carbocysteine increases chloride transport across the airway epithelium, which may also contribute to its mucoregulatory action [15].

## 4. Pharmacokinetics

Carbocysteine is available in two forms for the oral preparation: it can be presented conjugated to its lysine salt (SCMC–Lys), or in its active form (SCMC), where the lysine group is cleaved during gastric absorption. This molecule is rapidly absorbed after oral administration, achieving its peak serum concentration after 1 to 2 h (h), and its plasma half-life is 1 h 33 min [16]. Carbocysteine appears to penetrate lung tissue and respiratory mucus, suggesting local action [17,18], and its kinetics fit a one-compartment open model [16].

The metabolism of carbocysteine is known to be especially complex, as several pathways—such as decarboxylation, N-acetylation, sulfoxidation, deamination\transamination, and ester glucuronidation—are involved to different degrees. Approximately 30% to 60% of the drug is excreted in urine without being metabolized [14,19].

Carbocysteine resets the balance between sialomucins and fucomucins by increasing sialomucin concentrations and decreasing fucomucin levels. Moreover, it enhances chloride transport across the airway epithelium. Therefore, initially, carbocysteine was introduced into clinical practice as only a mucoregulatory agent. However, a substantial number of studies carried out in recent years have better characterized the mechanism of action of carbocysteine, highlighting a series of useful effects in the management of COPD. These studies allow us to consider carbocysteine as a drug with several actions useful in treating a wide spectrum of acute and chronic diseases of the respiratory tract.

The contribution of carbocysteine in reducing exacerbations observed in COPD patients occurs via multiple mechanisms:(1)Modulating mucins and ciliary functions;(2)Counteracting viral infections;(3)Counteracting bacterial infections;(4)Counteracting oxidative stress and exerting cytoprotective effects;(5)Improving steroid responsiveness;(6)Exerting anti-inflammatory activities.

### 4.1. Effects of Carbocysteine on Mucins and Ciliary Functions

Goblet cell hyperplasia and metaplasia are frequently found in the airway epithelium of COPD patients and some more severe asthma patients. These events cause changes in the glycoprotein components present in the mucus, altering its viscous properties [20]. Glycosidases or glycosyltransferases (e.g., fucosidase, sialidase, fucosyltransferase, and sialyltransferase) are responsible for these changes observed in patients. The polymeric mucins MUC5AC and MUC5B are integral components of airway mucus. MUC5AC is localized in goblet cells present at the surface epithelium and in the terminal secretory ducts of submucosal glands. MUC5AC secretion, in addition to airway smooth-muscle contraction, is necessary for airway hyper-responsiveness (AHR), and appears to be detrimental in acute lung injury, enhancing neutrophil trafficking and inflammation, along with the severity and abundance of mucus plugging. The MUC5B protein is localized in the mucous cells of submucosal glands and, to a lesser extent, in secretory cells within the surface airway epithelium of the trachea and bronchi. MUC5B is critical for mucociliary clearance and airway defense. Muc5b-deficient mice accumulate aspirated materials in the airways and develop chronic bacterial infections, severe inflammation, and airway obstruction.

Carbocysteine can increase cilia beating in airway epithelial cells, thus improving the function of the mucociliary escalator and its function of removing harmful particles, viruses, and bacteria from the airway surface [21].

In models employing sulfur dioxide (SO_2_) to induce airway insult, carbocysteine can counteract the enhanced inflammatory cell infiltration as well as increased mucus cell numbers and MUC5AC protein expression [22]. Furthermore, it has been demonstrated that carbocysteine has the ability to rebalance the altered glycosidase or glycosyltransferase activities, increase the levels of the MUC5AC mRNA and protein via repeated SO_2_ exposure in a rat model [23], and restore the physiological sugar composition of mucus by regulating fucose and sialic acid contents in mucins [24]. Carbocysteine also preserves this regulatory function in mucus composition in pathological models, where mucus viscosity alteration is promoted by tumor necrosis factor (TNF) [24].

### 4.2. Effects of Carbocysteine on Viral Infections

Viruses are detected in half of all COPD exacerbations, and are associated with poorer clinical outcomes. Rhinovirus, respiratory syncytial virus (RSV), and influenza are the most commonly detected viruses during exacerbation. Moreover, viral pathogens may play an important role in driving the progression of COPD by acting as triggers for exacerbation and subsequent decline in lung function [25]. In this regard, it has been demonstrated that carbocysteine pretreatment reduces the expression of the rhinovirus receptor intercellular adhesion molecule 1 (ICAM-1), as well as reducing viral titers and viral RNA in vitro using human tracheal epithelial cells [26]. Carbocysteine is also able to reduce the levels of cytokines (e.g., interleukin (IL)-6, IL-8, IL-1) released in tracheal epithelial cells infected with human RSV [27].

Yamaya et al., using an in vitro model of the airway (human trachea) represented by epithelial cells infected with the FluA virus, evaluated the efficacy of carbocysteine as an inhibitor of this virus. Interestingly, carbocysteine can reduce viral titers, RNA levels from the influenza A virus, and pro-inflammatory signals, including of the IL-6 cytokine and nuclear expression of the pro-inflammatory transcription factor nuclear factor kappa B (NF-κB). Moreover, it diminishes sialic acid with α2,6 linkage—a receptor for the influenza virus [28]. Considering that there are currently few specific and effective antiviral strategies for patients with COPD, carbocysteine may contribute to managing viral infections. In addition, COVID-19 patients show increased oxidative stress and uncontrolled pulmonary and systemic inflammation. N-acetylcysteine, carbocysteine, and erdosteine, as thiol agents, exert multiple activities that may hamper COVID-19 infection. Thiols block the angiotensin-converting enzyme 2, thereby counteracting the penetration of SARS-CoV-2 into the airway epithelia, or exert a broad range of antioxidant and anti-inflammatory mechanisms that mitigate pro-inflammatory responses and limit tissue damage. Although some data on the clinical efficacy of N-acetylcysteine as additional therapy in COVID-19 patients are emerging, limited data are available for carbocysteine or erdosteine. Future dedicated studies are warranted for assessing the efficacy of thiol treatments in COVID-19 patients.

### 4.3. Effects of Carbocysteine on Bacterial Infections

The adhesion of bacteria to pharyngeal epithelial cells is the initial step in the pathogenesis of bacterial infections. Disruption of bacterium–host cell interactions with receptor antagonists or modulation of the ensuing inflammatory profile can present attractive avenues for therapeutic development. Carbocysteine can reduce the attachment of *Moraxella catarrhalis*, non-typeable *Haemophilus influenzae* [29,30], and *Streptococcus* (*S.*) *pneumonia* [31]. Carbocysteine is also able to induce the detachment of already-adherent bacteria from epithelial cells [31]. While pro-inflammatory responses upregulate the expression of platelet-activating factor receptor (PAFR)—a receptor used by *S. pneumonia* to firmly adhere to epithelial cells—carbocysteine can reduce PAFR mRNA and protein levels, thus also decreasing bacterial adherence in a pro-inflammatory microenvironment [32]. Moreover, carbocysteine allows antibiotics to penetrate more easily through the hematobronchial barrier. The association of amoxicillin and carbocysteine leads to an enhancement in antibiotic levels in the bronchial secretion (even if it is purulent), performing a sterilizing action in a short timeframe, imparting a significant therapeutic advantage [33].

### 4.4. Antioxidant and Cytoprotective Effects of Carbocysteine

Oxidative stress induced by chronic smoke exposure is considered to be a crucial event in the pathogenesis of COPD [34]. Increased oxidative burden and antioxidant imbalance generated by inhaled oxidants or by endogenous sources are involved in cellular and tissue damage. For this purpose, strategies aimed at reducing oxidative burden or rising antioxidants are central in the therapy of COPD.

Different cellular models with increased oxidative stress are generated using hydrogen peroxide (H_2_O_2_), hypochlorous acid (HOCl), hydroxyl radicals (•OH), or cigarette smoke extract (CSE). Using an experimental method with •OH to enhance oxidative stress, Garavaglia et al. demonstrated that carbocysteine exerts scavenging activity, restoring physiological levels of reactive oxygen species (ROS) and preventing the irreversible reduction of both glutathione (GSH) and chloride currents generated by •OH exposure [35]. Furthermore, Guizzardi et al. demonstrated that the rebalancing effects of carbocysteine on the GSH levels were dependent on the correct functionality of the cystic fibrosis transmembrane regulator (CFTR) [36]. For this reason, this efflux of GSH was not observed in diseases such as cystic fibrosis, in which a correct function of CFTR is lacking.

The antioxidant activity of carbocysteine is strictly linked to cytoprotective and anti-inflammatory activities, since it (a) reverses the inactivation of α1-antitrypsin, and (b) reduces the higher production of IL-8 due to increased intracellular •OH activity [37]. Bazzini et al. showed that CSE significantly promotes cell mortality in a time- and dose-dependent manner, via an apoptosis-independent pathway. Short-term CSE exposure induces an increase in ROS levels and a reduction in intracellular GSH concentration. In parallel, the expression of glutathione peroxidases 2 and 3, glutathione reductase, and glutamate–cysteine ligase is increased. Carbocysteine is effective in counteracting all of these effects [38]. With regard to cell apoptosis, Yoshida et al. demonstrated that carbocysteine reduces the increased cell apoptosis (caspase (CASP)3 and CASP9 activation) in tracheal epithelial cells exposed to H_2_O_2_ [39]. Hanaoka et al. showed that carbocysteine reduces lung emphysema (alveolar enlargement and parenchymal destruction) caused by cigarette smoke exposure in rats. Carbocysteine preserves pulmonary and systemic antioxidant mechanisms, and diminishes both cell apoptosis and matrix metalloproteinase (MMP)-2 and -9 activity [40].

Evidence supports a pro-senescence role of increased oxidative stress. In this regard, Pace et al. showed the role played by CSE not only in decreasing cell proliferation, but also in promoting a senescent functional profile of bronchial epithelial cells. Thus, the nuclear expression and activity of sirtuin 1 (SIRT1) and forkhead box O3 (FoxO3) are reduced, whereas beta-galactosidase staining increases. In CSE-stimulated bronchial epithelial cells, carbocysteine reverses all of these phenomena by inducing cell proliferation, increasing SIRT1 and FoxO3 nuclear expression, and by lowering beta-galactosidase staining [41].

### 4.5. Effects of Carbocysteine on Steroid Responsiveness

The inflammatory events present in COPD or more severe asthma patients in stable conditions are often resistant to corticosteroids [42]. Oxidative stress contributes to the low response rate to corticosteroids through the downregulation of histone deacetylase (HDAC) activity—a crucial event in the transrepression activities of corticosteroids [43]. HDACs—deacetylating histones that bind to inflammatory gene promoters—limit the access of the transcriptional machinery to these genes, thus repressing their transcription [44]. CSE exposure causes marked pathological features of COPD that are insensitive to dexamethasone and are associated with downregulation of HDAC2 activity. Song et al. reported that increased oxidative stress contributes to steroid insensitivity in COPD or more severe asthma patients [45]. These authors provide evidence that, in both in vitro and in vivo animal models of oxidative-stress-mediated steroid insensitivity, carbocysteine restores oxidative stress levels by reducing ROS and promoting GSH and superoxide dismutase (SOD) activity [46]. The increased thiol–GSH levels due to carbocysteine increase HDAC2 recruitment and suppress H4 acetylation of the IL-8 promoter, potentiating dexamethasone activity even in high-oxidative-stress models.

The dexamethasone insensitivity observed in COPD rat models is improved by carbocysteine. Carbocysteine is crucial in inhibiting chronic lung inflammation (total and differential inflammatory cell counts, inflammatory cytokine release, and inflammatory cell infiltration) and ameliorating airway remodeling (thickness of the airway epithelium and smooth muscle and airway fibrosis). Moreover, it can improve emphysema (emphysema index D2, levels of MMP-9 in bronchoalveolar lavage fluid (BALF), and the expression of α-1 antitrypsin) and prevent impairments of lung function (e.g., PEF, intratracheal pressure (IP), and IP-slope). In an in vitro model of bronchial epithelial cells, carbocysteine, unlike fluticasone propionate, could counteract CSE-mediated effects, reducing oxidative stress as well as increasing HDAC2 levels [47]. These observations were further corroborated in a more recent paper, in which it was shown that CSE decreases HDAC3 expression and, consequently, increases acetylation processes within the cells. The improvement in acetylation promotes the expression of the pro-inflammatory transcription factor p-cAMP-responsive element-binding protein-1 (p-CREB), IL-1 mRNA, and neutrophil chemotaxis [48]. Instead, as CSE-exposed cells are incubated with carbocysteine and beclomethasone, deacetylation processes are induced, and HDAC3 inhibits p-CREB and IL-1 mRNA expression levels, as well as neutrophil chemotaxis.

### 4.6. Anti-Inflammatory Activities of Carbocysteine

A key component of innate immunity and innate defense mechanisms is represented by the Toll-like receptor (TLR) family [49]. TLRs, predominantly expressed by monocytes/macrophages and neutrophils [49], have also been identified in pulmonary and bronchial epithelial cells [50].

The airway epithelium is emerging as a regulator of innate immune responses to a variety of insults, including cigarette smoke. Products derived from epithelial cell injury can act as ligands for TLR4 and TLR2, thus amplifying inflammatory responses within the airways. In this regard, it has been demonstrated that upon CSE stimulation, the airway epithelium can release increased concentrations of IL-8, which sustain the influx of neutrophils into the airways, consequently triggering innate immune responses [50]. Neutrophils represent the predominant cell type in COPD patients. Neutrophil elastase (NE) promotes mucous cell metaplasia in chronic bronchitis, and is actively involved in the protease/anti-protease imbalance—a phenomenon that leads to lung tissue destruction and emphysema [4,51]. In vitro and in vivo studies provide evidence for the anti-inflammatory activities of carbocysteine on the abovementioned pro-inflammatory mechanisms.

## 5. In Vitro and In Vivo Studies of Carbocysteine’s Effects

### 5.1. In Vitro Studies of Carbocysteine’s Effects on Different Cell Lines

Carbocysteine counteracts some pro-inflammatory CSE-mediated effects in a human bronchial epithelial cell line (16-HBE). In CSE-stimulated bronchial epithelial cells, carbocysteine has been demonstrated to play a crucial role in the reduction in TLR4 expression and lipopolysaccharide (LPS) binding, IL-8 mRNA, and IL-8 release due to IL-1 stimulation and neutrophil chemotactic activity [52]. Similar anti-inflammatory effects of carbocysteine were observed in nasal epithelial cells exposed to CSE [53], providing compelling evidence that carbocysteine may also be considered a promising therapeutic strategy in chronic inflammatory nasal diseases. The effects of carbocysteine on neutrophils have also been confirmed by other observations. Ishii et al. showed that carbocysteine can reduce neutrophil chemotaxis not only by inhibiting N-formylmethionyl-leucyl-phenylalanine (fMLP)-mediated neutrophil adherence to pulmonary vascular endothelial cells, but also by decreasing the production of inositol 1,4,5-triphosphate (IP3) and diacylglycerol in neutrophils [54]. Other than reducing neutrophil chemotactic molecule expression/release, adhesion, and chemotaxis of neutrophils, Nogawa et al. stated that carbocysteine decreases rat neutrophils. Moreover, it shows scavenging effects on H_2_O_2_, HOCl, •OH, and peroxynitrite (ONOO-), thus reducing further pro-inflammatory responses (e.g., IL-8 and IL-6 release) [55]. In vitro models of distal airways (A549 cells) cultured with or without H_2_O_2_ [56] provide evidence that carbocysteine counteracts the effects of H_2_O_2_ by increasing cell viability, decreasing lactate dehydrogenase (LDH), IL-6, and IL-8 in cell supernatants, and attenuating the activation of extracellular signal-regulated kinase 1/2 (ERK1/2) and NF-κB. In the same in vitro model, Wang et al. demonstrated that carbocysteine, administered to the cells 24 h before or after TNF stimulation, can regulate the release of IL-6 and IL-8, as well as the activation of ERK1/2 and NF-κB [57]. The main molecular events promoted in vitro in airway cellular models by carbocysteine are reported in Figure 3.

### 5.2. In Vivo Studies of Carbocysteine’s Effects in Animal Models

Asti et al. demonstrated the efficacy of carbocysteine administered either orally or by inhalation to reduce airway hyper-reactivity and inflammation promoted by smoke exposure, using in vivo models [58]. In mice treated with intratracheal instillation of LPS, carbocysteine decreased neutrophil numbers by increasing the binding of apoptotic neutrophils to alveolar macrophages, and by promoting phagocytosis of neutrophils [59]. Song generated a murine model of COPD by instilling LPS and cigarette smoke exposure to study carbocysteine’s effects. He stated that carbocysteine significantly restored the MUC5B/MUC5AC ratio, together with decreased neutrophil counts, keratocyte-derived cytokine and IL-6 levels, and TNF-α mRNA expression in the studied mice. Furthermore, carbocysteine significantly improved lung function, as reflected by airway resistance and dynamic compliance [60].

## 6. In Vivo Studies of Carbocysteine’s Effects in COPD Patients

Limited evidence demonstrates the in vivo anti-inflammatory or antioxidant activities of carbocysteine in COPD or asthma patients (Figure 4).

In this regard, Carpagnano et al. measured 8-isoprostane and IL-6 levels in the exhaled breath condensates of mild acute and mild stable COPD patients, and demonstrated that carbocysteine can decrease the high concentrations of these mediators present in such patients [61]. Furthermore, a recent study provided data on short-term treatment (20 days) with or without carbocysteine to manage mild AECOPD, demonstrating that the addition of carbocysteine in mild AECOPD patients improves symptoms, forced expiratory volume in the 1st second (FEV_1_), and forced expiratory flow at 25–75% of forced vital capacity (FEF25-75), while it increases circulating sRAGE and reduces miR-21, IL-8, and fAGEs [62].

### Clinical Studies of Carbocysteine’s Effects in COPD

To date, few clinical studies regarding carbocysteine’s effects in COPD patients are available. In the UK, before 1990, small-scale studies investigated the effects of daily administration of carbocysteine (2.25–3 g) versus (vs.) placebo in chronic bronchitis patients. The results were rather heterogeneous in terms of FEV_1_ and peak flow rate, but the outcomes regarding more subjective parameters—such as cough, dyspnea, and increased sputum clearance—were consistently positive [63,64,65,66] (Figure 5 and Table 1).

An earlier, more accurate double–blind, parallel-group study developed in the UK compared the treatment effects after administration of 750 mg of carbocysteine three times daily vs. placebo in 109 patients with chronic bronchitis over 6 winter months. The results suggested no significant differences in the exacerbation rate, and an increase in the peak flow from baseline in both the placebo and intervention groups [67] (Figure 5 and Table 1). On the other hand, two RCTs conducted in Tokyo compared 1.5 g of daily carbocysteine with placebo in 156 COPD patients over 12 months. This study disclosed a significant reduction in the number of common colds, along with a lower exacerbation rate in COPD patients [26]. None of the patients received ICSs or oral corticosteroids during the study. Moreover, an Italian multicenter, prospective, double-blind RCT involving 662 COPD patients reported no significant differences in baseline FEV_1_, but a significantly prolonged mean time until the first exacerbation episode in patients treated with 2.7 g of SCMC–Lys once daily for 6 months compared to placebo [68] (Figure 5 and Table 1).

The PEACE study was the first clinical trial to closely follow a reliable research design to clarify whether COPD patients could benefit from protracted mucolytic therapy [69]. This study was classified as a multicenter (22 centers in China), randomized, double-blind, placebo-controlled, parallel-group study. To be enrolled in this study, patients had to have a history of at least two COPD exacerbations within the previous 2 years, and to be considered clinically stable for over 4 weeks before the study. The PEACE study enrolled more than 700 COPD patients, who were followed for 1 year. The cumulative number of exacerbations for the whole year was 325 in the carbocysteine group and 439 in the placebo group, corresponding to 1.01 (SE 0.06) exacerbations per patient/year with carbocysteine treatment vs. 1.35 (SE 0.06) with placebo. The risk ratio of exacerbation was 0.75 (95% CI 0.62–0.92, *p* = 0.004). Interestingly, the number of acute exacerbations in COPD patients who received carbocysteine (1500 mg/day for a year) decreased by 24% compared to the placebo group. Moreover, carbocysteine demonstrated a better rate of prevention in patients who suffered frequent exacerbation events, but these preventive effects were found only in patients treated with carbocysteine for 6 months or more [69] (Figure 5 and Table 1).

These results may lead to the conclusion that the anti-inflammatory and antioxidant properties of carbocysteine in COPD patients require time to be effective. Thus, the longer the carbocysteine administration, the better the preventive effects against recurrent exacerbation. Non-significant interaction was found between the preventive effects and COPD severity, smoking, and concomitant therapy. The advantage of carbocysteine over placebo in the prevention of exacerbations was noteworthy even after adjustment for concomitant use of ICSs. This finding has been to the object of debate, particularly because it differs from the conclusions of the BRONCUS study [70], in which a significant reduction in exacerbation rate by N-acetylcysteine was shown only in patients without concomitant use of ICSs. The authors of the PEACE study themselves provide some reasonable explanations for such differences. First of all, the response to treatment could be associated with patients’ ethnicity, as the PEACE study’s patients were Chinese. Secondly, PEACE patients were receiving ICSs at a lower percentage and a smaller dose compared with BRONCUS participants. As such, the effects of carbocysteine would be more readily identified in Chinese patients with modest use of concomitant ICSs, as opposed to the BRONCUS study.

**Figure 5 pharmaceutics-14-01261-f005:**
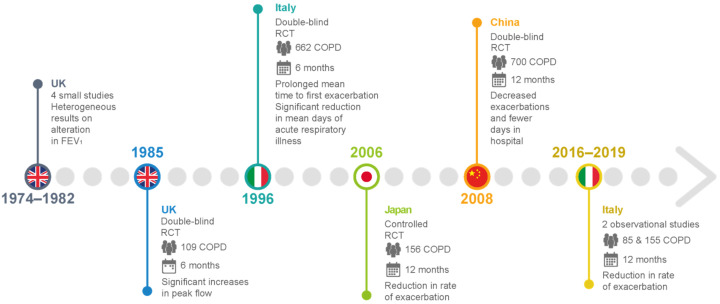
Main clinical studies of carbocysteine’s effects in COPD patients [10,26,63,64,65,66,67,68,69,71].

Mucolytics such as carbocysteine seem unlikely to replace ICSs in the treatment of COPD, but might be an important alternative where corticosteroid use is contraindicated. This study also documented improvements in St George’s Respiratory Questionnaire (SGRQ) total score and symptom score that were observed as clinically relevant. The nature and incidence of adverse events were similar to previous studies, and did not present differences between the carbocysteine and placebo groups for a 1-year treatment protocol, confirming carbocysteine’s good tolerability for long-term treatment.

More recently, the observational and prospective CAPRI study included 85 COPD patients with a history of at least one COPD exacerbation within the previous year. Enrolled patients were treated with daily administration of 2.7 g of SCMC–Lys, and examined every three months until the end of the study (i.e., 1 year). The primary endpoint—the reduction in the exacerbation rate—was reached after 12 months of therapy, whereas the PEACE study achieved the same result in 6 months. Interestingly, the decrease in the exacerbation frequency was completely independent of the use of ICSs, in contrast with the BRONCUS study [10]. The improvement in the quality of life was assessed as one of the secondary endpoints. The results showed an improvement in this parameter through a statistically significant decrease in SGRQ score, according to the PEACE study. The authors did not record a significant improvement in lung function (FEV1, FVC, FEV1/FVC). The BODE index was another parameter analyzed. In more detail, the BODE index is a multidimensional index that integrates body mass index (B), the degree of airflow obstruction (O), dyspnea (D), and exercise capacity (E), assessed by 6-min walking distance (6MWT) [71]. The BODE index is the best predictive parameter (with respect to FEV_1_) for estimating the risk of mortality in patients with COPD from both general and respiratory causes. In this study, the BODE index showed promising results, as it was significantly reduced, and was correlated with a significant improvement in the 6 MWT results.

**Table 1 pharmaceutics-14-01261-t001:** Clinical studies on the efficacy of carbocysteine treatment in COPD patients.

Clinical Trial	Study Design and Enrolled Patients	Therapeutic Regimen	Outcome
[63,64,65,66]	Small-scale UK studies on patients with chronic bronchitis	2.25–3.00 g carbocysteine daily vs. placebo	Heterogeneous results for alterations in FEV_1_, peak flow rate, and dyspnea scores
[67]	A double-blind, parallel-group study in the UK of 109 patients with chronic bronchitis over 6 winter months	750 mg carbocysteine three times daily compared with placebo in terms of peak flow and exacerbation rate.	No significant difference in exacerbation rate. Significant increases in peak flow from baseline in both placebo and intervention groups
[26]	Placebo-controlled RCTs in Tokyo for 156 patients with COPD over 12 months	1.5 g carbocysteine daily with placebo	Significant reduction in the number of common colds and reduction in the rate of exacerbation
[68]	An Italian multicenter, prospective, double-blind RCT involving 662 outpatients with chronic bronchitis	2.7 g SCMC–Lys once daily for 6 months in COPD patients	No significant difference in baseline FEV_1_ between the groups. Mean time to first exacerbation was significantly prolonged, and significant reduction in mean days of acute respiratory illness per patient.
[69]	Multicenter, randomized, double-blind, placebo-controlled, parallel-group study in China involving more than 700 COPD patients with a history of at least two COPD exacerbations within the previous 2 years	1500 mg/day carbocysteine for one year	Long-term (one year) use of carbocysteine produced a reduction in the numbers of exacerbations in patients with COPD. Decreased exacerbations and fewer days in the hospital. No loss of lung function, and improvement in health-related quality of life.
[10]	An Italian observational and prospective study including 85 COPD outpatients with a history of at least 1 COPD exacerbation within the previous year.	2.7 g of carbocysteine daily for one year	Reduction in the exacerbation rate after 12 months of therapy, completely independent of the use of ICSs. Statistically significant improvement in the quality of life assessed (decrease in SGRQ score) and the distance walked (6MWT), with a significant reduction in the BODE index. No significant improvement in lung function (FEV1, FVC, FEV1/FVC).
[72]	Observational prospective study of 155 consecutively enrolled COPD patients with a history of at least 1 COPD exacerbation within the previous year.	2.7 g of carbocysteine daily for one year	Reduction in the number of exacerbations at 1-year evaluation, irrespective of treatment with or without ICSs.

Finally, the results of the CAPRI study were confirmed by Paone et al. By enrolling 155 COPD patients, they demonstrated that the addition of a single dose of carbocysteine lysine salt to background therapy was able to significantly reduce the 1-year exacerbation rate. Notably, as for CAPRI’s primary outcome, this decrease was completely independent of the use of inhaled steroids [72] (Figure 5 and Table 1).

In 2019, the Cochrane meta-analysis for long-term use of mucolytic agents established a moderate statistically significant reduction in the number of exacerbations per patient [2]. This meta-analysis included 30 placebo-controlled, randomized clinical trials (RCTs), and involved a total of 10,377 patients. All trials had a randomized, double-blind, parallel-group design. Out of those 30 studies, only 5 included in the meta-analysis concerned carbocysteine, and only 1 specifically evaluated the long-term use of carbocysteine [73]. From the four studies selected, only the PEACE study was defined as high-quality.

The analysis involved 1357 patients treated for 10.4 months (range 6–12 months) with 1500 mg of carbocysteine daily. The results showed that long-term use of carbocysteine reduced the number of exacerbations by 0.43 per participant per year (95% CI, −0.57, −0.29), and decreased the number of patients with at least one exacerbation. Moreover, the long-term use of carbocysteine in patients with COPD may improve their quality of life, and the treatment was well-tolerated. Unfortunately, substantial heterogeneity was detected in the outcomes. This heterogeneity decreased when the analysis was limited to populations of non-Chinese ethnicity. However, the notion that ethnicity could contribute to the heterogeneity observed is still under scrutiny, and remains a controversial question.

## 7. Limitations and Pitfalls in Carbocysteine Studies

Clinical studies performed in COPD patients to check the effects of carbocysteine have produced heterogeneous results in terms of several clinical outcomes, including the exacerbation rate and lung function parameters. This heterogeneity has also been seen in further studies analyzing other mucolytic agents. The GOLD 2021 Guidelines limit regular therapy with mucolytics such as carbocysteine for COPD patients not treated with inhaled corticosteroids [13]. The data currently available on mucolytics do not allow the precise identification of the potential “target” population. This limitation is related to the heterogeneity of the patients included in the studies, to the different dosage regimens, and to the presence of different concomitant treatments. Future investigations of mucolytics that take into account the limitations of the previous studies are needed to better clarify the “target” population.

To date, no trial specifically focused on the effects of carbocysteine for long-term use has been conducted. Thus, the information available concerning the direct impact of carbocysteine on disease progression, survival, and efficacy in the event of simultaneous use of ICSs in COPD patients is very limited.

Esposito et al. and Paone et al. showed carbocysteine’s effectiveness in reducing the rate of COPD exacerbations independently from corticosteroid use. Indeed, they observed a significant reduction in the number of exacerbations in individuals with more than two exacerbations at baseline. It is important to emphasize that these studies have some pitfalls, in that they were a single-site, non-RCT studies, with a limited number of enrolled subjects; however, their observational design, and the applicability of the findings to general, real-life, clinical practice, may overcome these shortcomings.

Other well-conducted studies exploring these debated questions are therefore strongly desirable. The effect on the exacerbation rate appears to be related to the duration of the therapy. Currently, carbocysteine in COPD remains a well-tolerated treatment (meaning that it does not require discontinuation of the treatment) with a favorable safety profile that provides symptomatic relief to many patients with mucus production. Mild gastrointestinal effects (e.g., diarrhea, nausea, stomach upset) and headache occurred in about 15% of patients with bronchiectasis during a 3-month treatment with carbocysteine, and these side effects did not require discontinuation of the treatment [74].

## 8. Conclusions

Although carbocysteine has been considered for decades by many to be only a mucoactive drug, it exhibits a wide range of in vitro and in vivo anti-inflammatory and antioxidant activities, as demonstrated both in animal models and humans.

Carbocysteine may diminish inflammatory cell recruitment to the airways, attenuate endothelial injury and associated cough sensitivity, and act as a potent free radical scavenger. Taken together, these actions, in parallel with its mucoregulatory function, suggest that carbocysteine could play an important, beneficial role in the pathogenesis of COPD and other inflammatory airway diseases.

Its activity in several molecular mechanisms—including modulation of mucin and ciliary function, counteraction to oxidative stress, viral and bacterial infections, promotion of anti-inflammatory activities, and improvement in steroid responsiveness—forms the basis of the clinical activity of carbocysteine.

## Figures and Tables

**Figure 1 pharmaceutics-14-01261-f001:**
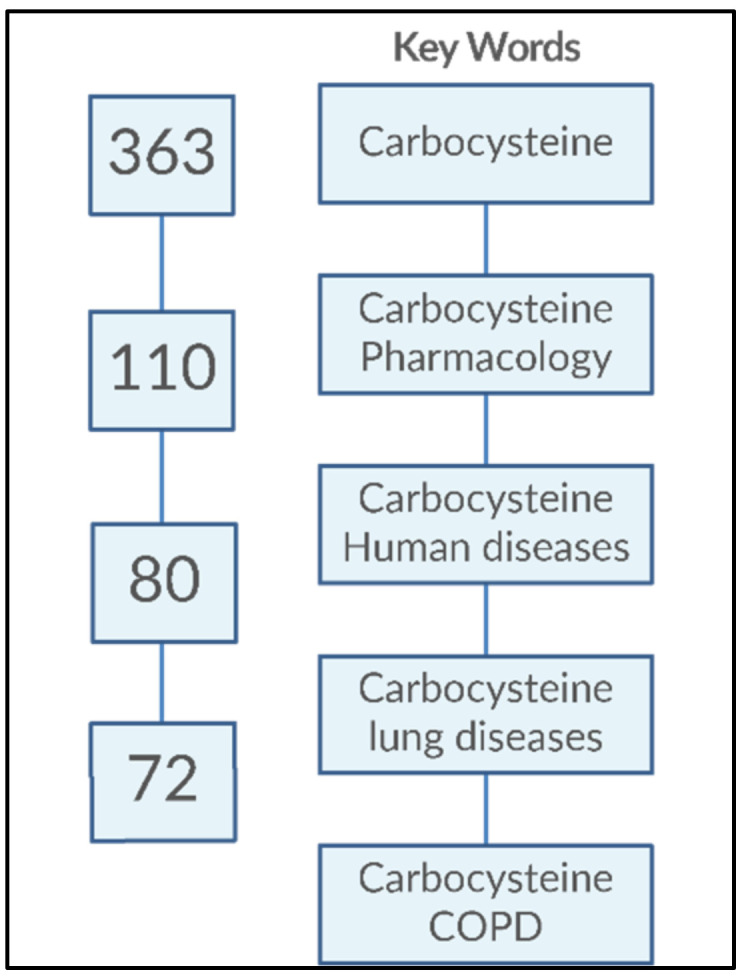
Flow diagram for article selection.

**Figure 2 pharmaceutics-14-01261-f002:**
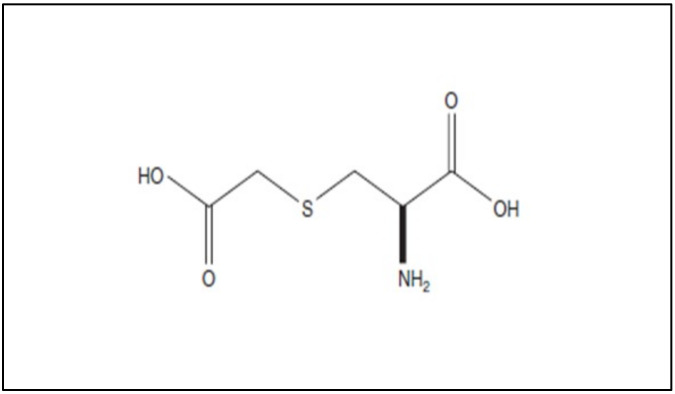
Chemical structure of carbocysteine.

**Figure 3 pharmaceutics-14-01261-f003:**
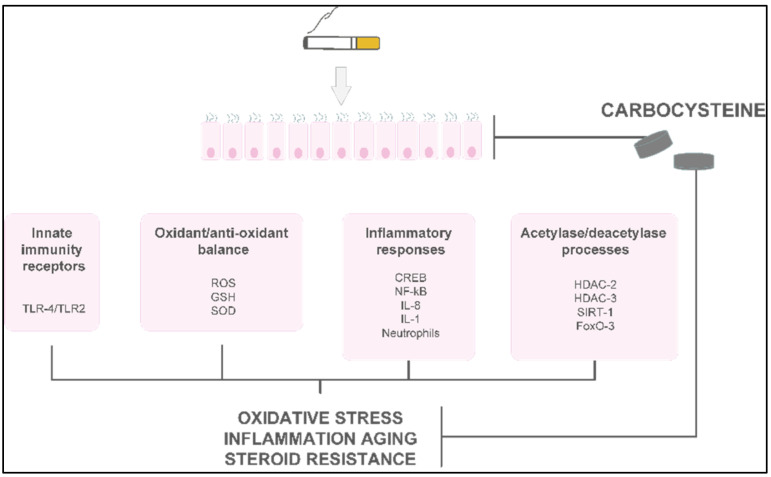
The main molecular mechanisms and effects of carbocysteine in smoke-injured airway epithelial cells.

**Figure 4 pharmaceutics-14-01261-f004:**
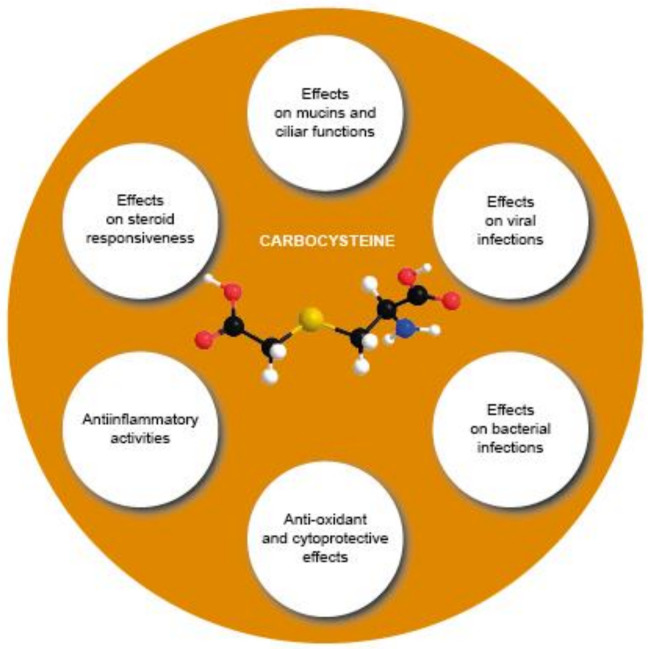
Overview of the various effects of carbocysteine with a positive impact in COPD.

## Data Availability

The data that support the findings of this study are available on request from the corresponding author, [FS].

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
