# Peer review of "Clinical Efficacy of Carbocysteine in COPD: Beyond the Mucolytic Action"

_pharmaceutics, 2022, doi:10.3390/pharmaceutics14061261_

Round 1
Reviewer 1 Report
The authors have written a nice review entitled "Clinical efficacy of carbocysteine in COPD: beyond the mucolytic action." This review is systematically organized and includes elements of basic and clinical research. There are some comments for this review.
- Line 59 incorrectly refers acute exacerbation of COPD as a subtype of COPD. Acute exacerbation is a status of COPD, not a subtype. Gold 2021 report lists the phenotypes of COPD. Please add a short summary of these phenotypes in introduction or discussion.
- Please add a schematic figure showing the molecular mechanisms/effects of carbocysteine
- This review discusses the role of carbocysteine and ICS use with details. However, some trials were performed more than one decade ago. Currently, ICS is less used in COPD. Please update the recommendation/indication of ICS use in COPD from Gold 2021 report in this review.
- In the era of COVID pandemic, COVID infection is a risk of acute exacerbation of COPD. Is there any data on mucolytic and antioxidant agents (N-acetylcysteine, carbocysteine, and erdosteine) and covid infection?
- There are limitations and pitfalls of previous studies of carbocysteine. Please write a paragraph to summary suggestions for future research design. The main issues include phenotypes of COPD, dose/duration of carbocysteine, biomarker of oxidative stress and inflammation, ethnicity, ICS use, clinical outcomes.
- The authors list the limitations of current studies. Here, GOLD recommendations that “mucolytic and antioxidant agents (N-acetylcysteine, carbocysteine, and erdosteine) to treat COPD patients that are not receiving inhaled corticosteroid therapy” should be pointed out. This has been previously described by the authors themselves.
- Line 221 FluA virus is not appropriate. Please replace FluA virus with influenza A virus
- Line 481 expand the description of BODE, including components, prognosis, reference
- Table 1 In outcome of China trial, please add the description that the AE reduction is with or without concomitant use of ICS”
- Figure 3, Table 1, Tokyo is the capital of Japan, may use the nation name “Japan” just like other trials (UK, China, Italy)
Author Response
All the revisions to the manuscript requested by the reviewer have been highlighted in yellow or in track changes.
- Line 59 incorrectly refers acute exacerbation of COPD as a subtype of COPD. Acute exacerbation is a status of COPD, not a subtype. Gold 2021 report lists the phenotypes of COPD. Please add a short summary of these phenotypes in introduction or discussion.
RESPONSE FROM AUTHORS
We really thank the reviewer for the general positive evaluation of our paper. Following the reviewer’s suggestion we have now modified the sentence in the revised version of the manuscript as follows: A subtype of COPD patients with frequent exacerbations, experienced the recurrence of episodes characterized by an acute worsening of respiratory symptoms that require additional therapy. Indeed, the recurrence of these episodes accelerates the loss of lung function and leads to decreased health-related quality of life, significant economic costs, and increased mortality rates.
- Please add a schematic figure showing the molecular mechanisms/effects of carbocysteine
RESPONSE FROM AUTHORS
We have now provided a schematic new figure (Fig. 3) that reports the main the molecular mechanisms/effects of carbocysteine
- This review discusses the role of carbocysteine and ICS use with details. However, some trials were performed more than one decade ago. Currently, ICS is less used in COPD. Please update the recommendation/indication of ICS use in COPD from Gold 2021 report in this review.
RESPONSE FROM AUTHORS
Following Reviewer’s suggestion, as reported in Gold 2021, we have now specified that ICS is strongly recommended as an add-on therapy for COPD patients who are highly symptomatic, have a severe exacerbation history and high eosinophil count or for patients where COPD and asthma can coexist.
- In the era of COVID pandemic, COVID infection is a risk of acute exacerbation of COPD. Is there any data on mucolytic and antioxidant agents (N-acetylcysteine, carbocysteine, and erdosteine) and covid infection?
RESPONSE FROM AUTHORS
We now added in the revised version of the manuscript a sentence on the potential role of thiol agents in COVID-19 patients.
COVID-19 patients in the most severe forms are characterized by increased oxidative stress and uncontrolled lung and systemic inflammation. N-acetylcysteine, carbocysteine, and erdosteine as thiol agents exert multiple activities that may hamper COVID-19 infection. Thiols block the angiotensin-converting enzyme 2 thereby counteracting penetration of SARS-CoV-2 into the airway epitheliums or exert broad range of antioxidant and anti-inflammatory mechanisms that mitigate pro-inflammatory responses and limit tissue damage. Although some data on clinical efficacy of N-acetylcysteine as add on therapy in COVID-19 patients are emerging, limited data are available for carbocysteine or erdosteine.
- There are limitations and pitfalls of previous studies of carbocysteine. Please write a paragraph to summary suggestions for future research design. The main issues include phenotypes of COPD, dose/duration of carbocysteine, biomarker of oxidative stress and inflammation, ethnicity, ICS use, clinical outcomes.
RESPONSE FROM AUTHORS
Following the reviewer’s suggestions we now included a paragraph named “Limitations and Pitfalls in carbocysteine studies”.
- The authors list the limitations of current studies. Here, GOLD recommendations that “mucolytic and antioxidant agents (N-acetylcysteine, carbocysteine, and erdosteine) to treat COPD patients that are not receiving inhaled corticosteroid therapy” should be pointed out. This has been previously described by the authors themselves.
RESPONSE FROM AUTHORS
Gold 2021 Gluidelines limit regular therapy with mucolytics such as carbocysteine to COPD patients not treated with inhaled corticosteroids. The data currently available on mycolitics do not allow the precise identification of the potential "target" population. This limitation is related to the heterogeneity of the patients included in the studies, to the different dosage regimens and to the presence of different concomitant treatments. Future studies on mucolytics that take into account the limitations of the previous studies could better clarify the "target" population for the correct allocation of mucolytics. These considerations were now added in the revised version of the paper.
- Line 221 FluA virus is not appropriate. Please replace FluA virus with influenza A virus
RESPONSE FROM AUTHORS
We have corrected the text as suggested.
- Line 481 expand the description of BODE, including components, prognosis, reference
RESPONSE FROM AUTHORS
As suggested by the reviewer we have now expanded the description of BODE.
- Table 1 In outcome of China trial, please add the description that the AE reduction is with or without concomitant use of ICS”
RESPONSE FROM AUTHORS
As reported in the Methods of the PEACE study, “Conventional treatment for COPD, such as short-acting or long-acting bronchodilators and inhaled corticosteroids, that had been started before the study, was permitted to continue but had to be sustained during the study period”. This statement doesn’t allow to precisely answer to the reviewer’s comment.
- Figure 3, Table 1, Tokyo is the capital of Japan, may use the nation name “Japan” just like other trials (UK, China, Italy)
RESPONSE FROM AUTHORS
We really apologize for the mistake. We have now included Japan and not Tokyo in the table 1.

Reviewer 2 Report
The authors of this manuscript (Manuscript ID: 1695610) provide an extensive coverage of the clinical efficacy of the mucoactive drug, carbocysteine. The authors are pointing out additional beneficial actions of the drug in the management of COPD along with the mucoregulatory function. However, the exact molecular mechanism of its action remains largely unknown and requires further investigation. Overall, the manuscript is well written, interesting and covers comprehensively the existing knowledge.
Author Response
All the revisions to the manuscript requested by the reviewer have been highlighted in yellow or in track changes.
The authors of this manuscript (Manuscript ID: 1695610) provide an extensive coverage of the clinical efficacy of the mucoactive drug, carbocysteine. The authors are pointing out additional beneficial actions of the drug in the management of COPD along with the mucoregulatory function. However, the exact molecular mechanism of its action remains largely unknown and requires further investigation. Overall, the manuscript is well written, interesting and covers comprehensively the existing knowledge.
RESPONSE FROM AUTHORS
We really thank the reviewer for the general positive evaluation of our paper. We have now provided a new figure (Fig.3) showing the molecular mechanisms/effects of carbocysteine.

Reviewer 3 Report
it was a pleasure reading your article. it is a very comprehensive and up-to-date review of carbocysteine and its many overlooked benefits. I would have some remarks:
introduction:
line 59 - AECOPD -"a subtype of COPD" -I feel it should be replaced, as every COPD subtype has exacerbations
line 111- the aim of this review should be more clearly stated
Material and methods:
a diagram may be useful as it could show how the remains article have been selected, according to which criteria
Body of the text
maybe a paragraph about the possible side effects of carbocisteine
Conclusions
should be a little more concise.
Author Response
All the revisions to the manuscript requested by the reviewer have been highlighted in yellow or in track changes.
Introduction:
line 59 - AECOPD -"a subtype of COPD" -I feel it should be replaced, as every COPD subtype has exacerbations
RESPONSE FROM AUTHORS
We really thank the reviewer for the general positive evaluation of our paper. Following the reviewer’s suggestion we have now modified the sentence in the revised version of the manuscript as follows: A subtype of COPD patients with frequent exacerbations, experienced the recurrence of episodes characterized by an acute worsening of respiratory symptoms that require additional therapy. Indeed, the recurrence of these episodes accelerate the loss of lung function and leads to decreased health-related quality of life, significant economic costs, and increased mortality rates.
line 111- the aim of this review should be more clearly stated
RESPONSE FROM AUTHORS
We have now better specified the aim of the review.
Material and methods:
a diagram may be useful as it could show how the remains article have been selected, according to which criteria
RESPONSE FROM AUTHORS
We have now included a new figure (Fig.1) with a diagram.
Body of the text
maybe a paragraph about the possible side effects of carbocysteine
RESPONSE FROM AUTHORS
We include a sentence on the side effects of carbocysteine.
Conclusions
should be a little more concise.
RESPONSE FROM AUTHORS
We have now revised the conclusions.

Author Response
All the revisions to the manuscript requested by the reviewer have been highlighted in yellow or in track changes.
RESPONSE FROM AUTHORS
The manuscript is a systematic review and we have specified in the main page.
As the reviewer suggested, more specific terms for literature search have been included in the manuscript.
Italics was used for “in vitro” term.
We corrected the citation indicated in the text changing it into “Carpagnano et al. measured 8-isoprostane and IL-6 levels in exhaled breath condensate of mild acute and mild stable COPD and demonstrated”.
Empty lines have been removed.
The last raw of the table was highlighted by we could not find any comment associated to the highlighted sentence.

Round 2
Reviewer 1 Report
In page 11, the old figure is not removed.
Good corrections.